# TEST-TIME ADAPTATION FOR UNSUPERVISED COMBINATORIAL OPTIMIZATION

## ABSTRACT

Neural combinatorial optimization (NCO) has emerged as a data-driven alternative to classical solvers, with recent advances in unsupervised learning (UL) frameworks enabling training without ground truth solutions. However, current UL-based NCO approaches tend to emphasize either generalization across diverse problem instances or instance-specific optimization. In this work, we introduce TACO, a model-agnostic test-time adaptation framework that unifies and extends these two paradigms through principled warm-starting: beginning from a trained, generalizable NCO model and applying instance-specific model updates. Crucially, compared to naively fine-tuning a trained generalizable model or optimizing an instance-specific model from scratch, TACO achieves better solution quality while incurring negligible additional computational cost. Our method integrates seamlessly into existing UL-based NCO pipelines. Experiments on canonical CO problems, Minimum Vertex Cover and Maximum Clique, demonstrate the effectiveness and robustness of TACO across static, distribution-shifted, and dynamic settings, establishing its broad applicability and practical impact.

## 1 INTRODUCTION

Combinatorial optimization (CO) problems are central to many real-world applications, ranging from routing and scheduling to resource allocation and logistics (Papadimitriou & Steiglitz, 1982). These problems are notoriously hard to solve at scale due to their discrete and often non-convex structure. Neural combinatorial optimization (NCO) has emerged as a promising alternative to traditional solvers by learning solution heuristics directly from data (Joshi et al., 2019; 2022; Gasse et al., 2019; Hudson et al., 2022; Bello et al., 2016; Khalil et al., 2017; Li et al., 2024; 2023). Recent advances in unsupervised learning (UL) frameworks have enabled the training of powerful solvers without requiring optimal or near-optimal solutions (Karalias & Loukas, 2020; Wang & Li, 2023; Toenshoff et al., 2021; Schuetz et al., 2022; Wang et al., 2022).

Two primary paradigms have emerged within UL-based NCO: *generalization*-focused and *instance-based* methods. The first focuses on learning problem-specific heuristics from a diverse set of training instances, aiming for strong *generalization* to unseen problem instances (Karalias & Loukas, 2020; Wang & Li, 2023). Once trained, these models are typically deployed to generate solutions in a single forward pass, with no feedback or adaptation to the specific test instance. While this allows for efficient inference, it limits performance in scenarios involving distribution shifts or dynamic constraints, common conditions in real-world applications (Yang et al., 2012; Zhang et al., 2021).

In contrast, the second paradigm focuses on *instance-specific* optimization, where a model is optimized independently for each test instance, aiming for instance-wise good solutions, without requiring access to a training dataset containing diverse graph structures (Schuetz et al., 2022; Ichikawa, 2024; Heydaribeni et al., 2024). As a result, this paradigm stays unaffected by distribution shifts and dynamic changes, but lacks the ability to generalize from broader patterns and is potentially susceptible to becoming trapped in poor local optima during optimization (Wang & Li, 2023; Liao et al., 2025).

In this work, we propose TACO (**T**est-time **A**daptation for unsupervised **C**ombinatorial **O**ptimization), a test-time adaptation framework that unifies and extends these two approaches to develop a method that can simultaneously learn from broader patterns, efficiently adapt to specific instances, and adjust to distribution shifts or dynamic environments when needed. We do so by framing the fusion of these paradigms as a *principled warm-starting* procedure: we begin from a trained NCO model

with generalizability and adapt it to each test instance via effective, instance-specific updates. This design leverages the generalization capabilities of models while introducing adaptability without the computational burden of training from scratch. We note that bridging the two existing paradigms is non-trivial: a straightforward way to combine these two approaches would be to simply fine-tune a trained generalizable model. However, we show that such a tuned model often underperforms a freshly initialized one optimized from scratch in an instance-specific manner. This is because the optimization landscape around trained parameters may be less conducive to rapid adaptation, potentially due to overfitting or local minima.

To address this challenge, we incorporate a structured warm-starting technique for neural network training, and show that compared to both naively fine-tuning a trained generalizable model and optimizing an instance-specific model from scratch, our test-time adaptation strategy consistently yields superior solution quality with negligible additional computational overhead. TACO achieves this by enabling more flexible and exploratory adaptation while still leveraging learned inductive biases. Our method is model-agnostic and complements existing UL-based NCO pipelines, offering plug-and-play integration. We evaluate TACO with two existing NCO frameworks as backbones across canonical CO tasks, Minimum Vertex Cover and Maximum Clique, under static settings, distribution shifts, and dynamic environments. The results demonstrate consistent performance improvements, underscoring the generality and practical utility of our approach.

## 2 Preliminaries

Let $G = (V, E)$ be an undirected graph, where $V$ is the set of nodes with $|V| = n$, and $E \subseteq V \times V$ is the set of edges. We define the solution to a CO problem on graph $G$ as a vector $x \in \mathcal{X}(G)$, where $\mathcal{X}(G) \subseteq \{0, 1\}^n$ denotes the feasible solution space over $G$, depending on the problem constraints. The general form of a CO problem can thus be written as:

$$\min_{x \in \mathcal{X}(G)} f(G, x),$$

where $f(G, x)$ is a problem-specific objective function, such as vertex cover size or negative clique size, and $\mathcal{X}(G)$ encodes constraints like covering or connectivity. We begin with an overview of the two existing paradigms for UL-based NCO.

### 2.1 Unsupervised NCO with generalization: Erdős Goes Neural (EGN) and Meta-EGN

**EGN.** To tackle this problem in a label-free setting, Karalias & Loukas (2020) introduced EGN, a principled unsupervised learning approach inspired by Erdős' probabilistic method. Concretely, a Graph Neural Network (GNN) $g_\theta$ is trained by minimizing the objective to map an input graph $G$ to a distribution $D = g_\theta(G)$ over binary vectors $x \in \{0, 1\}^n$. Each component $x_i$ is modeled as a Bernoulli random variable with probability $p_i = g_\theta(G)_i$, denoting the likelihood of including node $v_i$ in the solution. In the constrained setting, constraint violations are penalized by augmenting the objective function:

$$\ell(D; G) := \mathbb{E}_{x \sim D}[f(G, x)] + \beta \cdot \mathbb{P}(x \notin \mathcal{X}(G)), \tag{1}$$

where $\beta \in \mathbb{R}_{>0}$ is a penalty parameter. Once trained, the learned distribution is used to decode a discrete solution via sequential decoding. This sequential process greedily fixes binary decisions $x_i$, one node at a time, so that the assignment of that node maintains or improves the expected objective. This ensures a deterministic and constraint-valid binary solution.

**Meta-EGN.** While EGN learns generalizable heuristics from training data, it optimizes for averaged performance over the distribution of problem instances and may fail to provide high-quality solutions for individual test instances, especially under distribution shifts. To overcome this limitation, Wang & Li (2023) proposed Meta-EGN, a meta-learning extension of EGN designed to refine the model for improving instance-wise solutions. Inspired by Model-Agnostic Meta-Learning (MAML) (Finn et al., 2017), Meta-EGN views each training instance as a pseudo-test case. Instead of directly learning a solution-generating network, Meta-EGN seeks to learn a parameter initialization that can be quickly fine-tuned on unseen test instances. During inference, Meta-EGN either uses the pre-adapted model or performs gradient updates for further refinement.

Meta-EGN offers instance-wise adaptability by leveraging meta-learning *during training*. In contrast, TACO improves adaptability *at test-time*. We show that our approach of test-time adaptation at times outperforms meta-learning-based adaptation, and that the performance of Meta-EGN can be further improved with negligible additional overhead when the two approaches are paired.

## 2.2 Unsupervised NCO with instance-specific optimization: PI-GNN

PI-GNN (Schuetz et al., 2022) is a general UL framework for CO problems formulated as a quadratic unconstrained binary optimization (QUBO) (Lucas, 2014; Glover et al., 2018; Djidjev et al., 2018). Given an instance of a CO problem, PI-GNN learns a solution via $g_\theta(G)$. Since the input graph lacks node features, PI-GNN initializes learnable node embeddings randomly and passes them through a GNN. The model outputs a relaxed solution $x \in [0,1]^n$ by optimizing a differentiable QUBO objective, followed by a rounding step to produce a valid binary solution. As PI-GNN directly applies a GNN to each problem instance individually and optimizes the corresponding QUBO objective, it operates in a fully training-data-free, instance-specific manner.

## 3 Method

Unifying the strengths of generalization and instance-specific optimization in unsupervised NCO, our method, TACO (**T**est-time **A**daptation for unsupervised **C**ombinatorial **O**ptimization), builds upon trained unsupervised NCO models and adapts them to individual test instances through a principled warm-starting procedure. In this work, we instantiate our method using EGN and Meta-EGN as backbone architectures. Unlike prior instance-specific approaches that optimize from randomly initialized parameters at test time (e.g., PI-GNN), TACO treats adaptation as a structured warm-start problem, leveraging learned inductive biases for fast and effective instance-level refinement.

For each test instance, TACO performs a small number of unsupervised gradient updates starting from the trained parameters $\theta$, using the loss function of the same form employed during training, as defined in Equation 1. Crucially, instead of directly initializing from $\theta$, TACO applies a strategic initialization, to preserve learned inductive biases while enabling effective adaptation. Specifically, the adapted parameters are initialized as:

$$\theta^* \leftarrow \lambda_{\text{shrink}} \cdot \theta + \lambda_{\text{perturb}} \cdot \epsilon,$$

where $0 < \lambda_{\text{shrink}} < 1$, $0 < \lambda_{\text{perturb}} < 1$, and $\epsilon \sim \mathcal{N}(0, \sigma^2)$. Here, $\lambda_{\text{shrink}}$ contracts the parameters towards the origin, reducing model overconfidence and encouraging gradient diversity, while $\lambda_{\text{perturb}}$ introduces mild noise that facilitates exploration of nearby solutions. In practice, $\lambda_{\text{shrink}}$ and $\lambda_{\text{perturb}}$ can be selected using a validation set or the test instances available at hand.

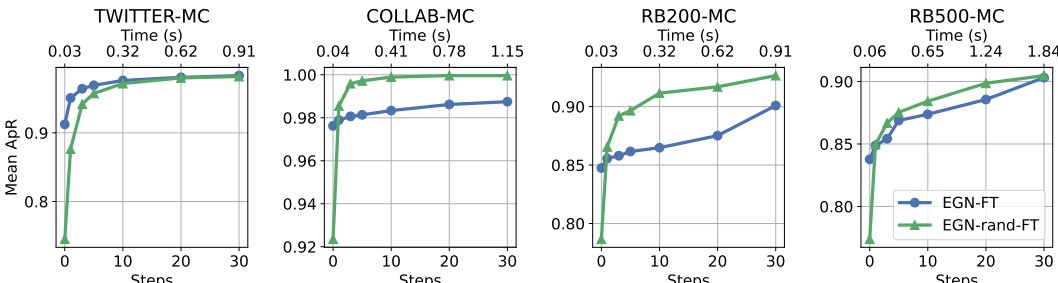

Figure 1: Performance ($\uparrow$) of trained and randomly initialized (rand) EGN models with respect to the number of fine-tuning (FT) steps. Detailed setup is explained in Section 4.

Originally proposed for a different problem of addressing the generalization gap in *supervised learning* tasks caused by naively warm-starting neural network training (Ash & Adams, 2020), shrink and perturb (SP) improves adaptation in our setting by (i) preserving the inductive bias encoded in the trained weights, (ii) accelerating convergence through a more favorable initialization, and (iii) helping escape poor local minima via stochastic perturbations. Notably, we observed that directly fine-tuning trained models at test time can often underperform freshly trained instance-specific models, even with identical update steps, as illustrated in Figure 1. This phenomenon suggests that the optimization

landscape around trained parameters may be less conducive to rapid adaptation, potentially due to overfitting or local minima. In contrast, the SP initialization used in TACO helps mitigate these issues by enabling more flexible and exploratory adaptation while still leveraging prior knowledge. Empirically, combining SP with trained NCO models for instance-wise adaptation yields consistently better performance than naive fine-tuning and optimization from scratch, as shown in Section 4.

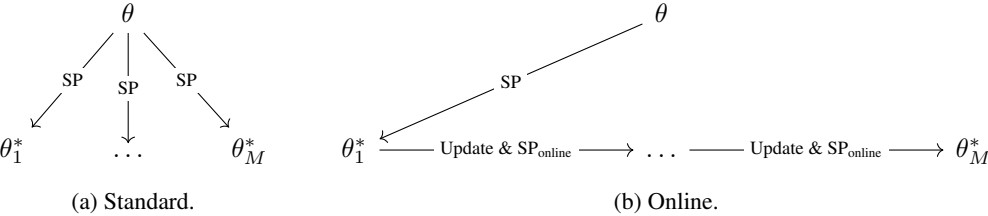

(a) Standard.                                    (b) Online.

Figure 2: Two versions of TACO: standard vs. online.

**Online TACO.** Given a sequence of test instances $\{G_1, G_2, \ldots, G_M\}$, standard TACO initializes each set of parameters from a fixed SP-transformed $\theta$. A natural extension is to make the process online: reusing the optimized parameters from instance $G_i$ as the initialization for instance $G_{i+1}$. That is, we warm-start from the most recent optimized $\theta_i^*$ and apply a fresh SP transformation with possibly different $\lambda_{\text{shrink}}$ and $\lambda_{\text{perturb}}$ before adaptation. This allows the model to accumulate knowledge across instances. We illustrate the differences between the standard and online variants in Figure 2.

## 4 EXPERIMENTS

We empirically evaluate the effectiveness of TACO and online TACO on classical CO problems defined over graphs: Minimum Vertex Cover (MVC) and Maximum Clique (MC). We consider three settings: (1) static graphs with fixed distribution, (2) distribution shifts in graph structures, and (3) dynamic graphs with temporal changes.

### 4.1 DATASETS

**Static problems.** For the static setting, we employed real-world and synthetically generated graphs used in previous works (Karalias & Loukas, 2020; Karalias et al., 2022; Wang & Li, 2023; Sanokowski et al., 2024). The real-world datasets include Twitter (Leskovec & Krevl, 2014) and COLLAB (Yanardag & Vishwanathan, 2015), which represent social and collaboration networks, respectively. Additionally, we generated synthetic graphs using the RB model (Xu et al., 2007), producing two datasets: RB200 and RB500, with approximately 200 and 500 nodes per graph. Following Wang & Li (2023), we sampled the RB model parameter $p \in [0.3, 1]$ uniformly when generating the training and validation sets and fixed $p = 0.25$ for the test set to generate hard instances. For Twitter and COLLAB, we used a standard 60-20-20 train/validation/test split. For RB200 and RB500, we generated 2000 graphs for training, 100 graphs for validation, and 100 graphs for testing.

**Distribution shift.** To assess performance under distribution shift, we trained our models on the Twitter dataset and evaluated them on the RB200 test set similar to Wang & Li (2023). This setup introduces a significant structural shift from real-world social graphs to synthetic rule-based graphs.

**Dynamic problems.** For the dynamic setting, we considered discrete-time dynamic graphs where a stream of graph snapshots is observed sequentially. Models were trained on static Twitter graphs and evaluated on two dynamic datasets: Twitter Tennis UO (Béres et al., 2018), a dynamic Twitter mention graph, for the MVC experiments, and COVID-19 England (Panagopoulos et al., 2021), a dynamic mobility graph, for the MC experiments. We took the top 150 popular nodes of Twitter Tennis UO for each snapshot, resulting in changes in both the node set $V$ and the edge set $E$. For COVID-19 England, the node set $V$ remains the same across all snapshots, and only the edge set $E$ changes over time. Both datasets are available in the PyTorch Geometric Temporal library (Rozemberczki et al., 2021). We selected Twitter Tennis UO for the MVC experiments only, since the clique sizes are in the range of 2 to 5, making performance comparison less meaningful. Similarly, COVID-19 England has vertex covers equal to the node set, so we used it for the MC experiments only.

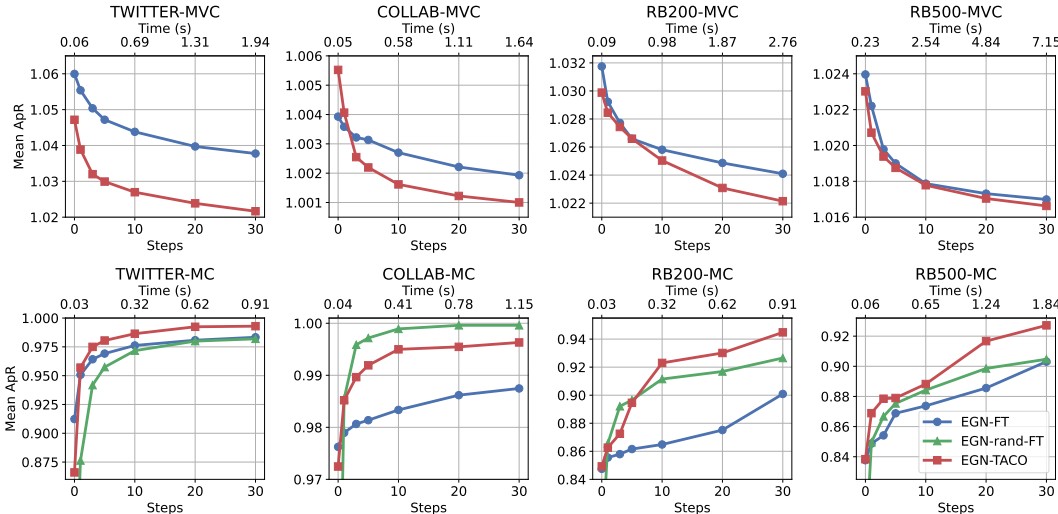

Figure 3: Mean ApR of methods using EGN as the backbone on static MVC (↓) and MC (↑) problems with respect to the number of update steps. "FT" stands for fine-tuning; "rand" means models are freshly initialized. The wall clock time factors in the decoding operations. Subplots not showing results of "EGN-rand-FT" are zoomed in for better illustration (i.e., freshly initialized models perform much worse). Figure 5 in Appendix B shows all results.

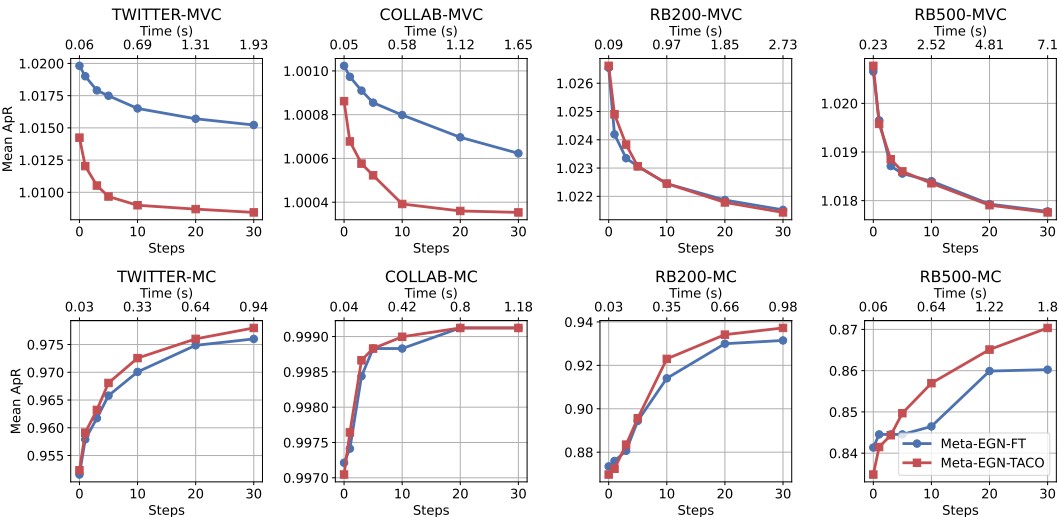

Figure 4: Mean ApR of methods using Meta-EGN as the backbone on static MVC (↓) and MC (↑) problems with respect to the number of update steps. "FT" stands for fine-tuning. The wall clock time factors in the decoding operations.

## 4.2 IMPLEMENTATION DETAILS AND BASELINES

Our EGN and Meta-EGN backbone models are the same as the ones in prior works (Karalias & Loukas, 2020; Wang & Li, 2023), consisting of four Graph Isomorphism Network (Xu et al., 2019) layers. We used the Adam optimizer (Kingma & Ba, 2015) for training and tuning all models. For evaluation, we obtained the ground truth for each graph (snapshot) by solving the corresponding CO problem using the Gurobi solver (Gurobi Optimization, LLC, 2024) and report the mean approximation ratio (ApR) as the primary metric. Additional implementation details, including the exact loss functions used for training and tuning all models, as well as the full set of optimizer, loss, and SP hyperparameters, are provided in Appendix A.

For baselines, we include fine-tuning trained and freshly initialized EGN and Meta-EGN models. We apply a number of unsupervised gradient updates and compare the best solutions achieved so far by the baseline methods and TACO. Since EGN and Meta-EGN take a random one-hot vector of the

Table 1: Mean ApR and seconds per graph of all methods on MVC ($\downarrow$). All results are from 30 update steps. "FT" stands for fine-tuning; "Accurate (8)" means 8 seeds were used. The best solutions are in **bold**, and the cases where EGN + TACO outperform Meta-EGN-FT(-Online) are in gray.

| TWITTER | | | |
|---|---|---|---|
| Method | Fast (1) | Balanced (4) | Accurate (8) |
| EGN | $1.09349_{\pm 0.05062}(0.02)$ | $1.05996_{\pm 0.03552}(0.06)$ | $1.04928_{\pm 0.02902}(0.13)$ |
| EGN-FT | $1.06311_{\pm 0.03905}(0.48)$ | $1.03775_{\pm 0.02640}(1.94)$ | $1.03026_{\pm 0.02123}(3.88)$ |
| EGN-FT-Online | $1.04482_{\pm 0.03484}(0.50)$ | $1.02434_{\pm 0.02197}(1.95)$ | $1.01865_{\pm 0.01669}(3.94)$ |
| EGN-TACO | $1.03826_{\pm 0.03473}(0.49)$ | $1.02165_{\pm 0.02221}(1.95)$ | $1.01668_{\pm 0.01636}(3.90)$ |
| EGN-TACO-Online | $\mathbf{1.03082_{\pm 0.03329}(0.50)}$ | $\mathbf{1.01997_{\pm 0.02094}(1.99)}$ | $\mathbf{1.01357_{\pm 0.01417}(3.96)}$ |
| Meta-EGN | $1.02998_{\pm 0.02016}(0.02)$ | $1.01981_{\pm 0.01579}(0.06)$ | $1.01749_{\pm 0.01536}(0.12)$ |
| Meta-EGN-FT | $1.02304_{\pm 0.01693}(0.48)$ | $1.01522_{\pm 0.01392}(1.93)$ | $1.01325_{\pm 0.01283}(3.87)$ |
| Meta-EGN-FT-Online | $1.02454_{\pm 0.01737}(0.49)$ | $1.01430_{\pm 0.01312}(1.95)$ | $1.01238_{\pm 0.01200}(3.90)$ |
| Meta-EGN-TACO | $\mathbf{1.01472_{\pm 0.01426}(0.49)}$ | $\mathbf{1.00844_{\pm 0.01004}(1.95)}$ | $\mathbf{1.00728_{\pm 0.00922}(3.91)}$ |
| Meta-EGN-TACO-Online | $1.02947_{\pm 0.03407}(0.50)$ | $1.01634_{\pm 0.01536}(1.96)$ | $1.01319_{\pm 0.01413}(3.93)$ |

| COLLAB | | | |
|---|---|---|---|
| Method | Fast (1) | Balanced (4) | Accurate (8) |
| EGN | $1.01197_{\pm 0.03309}(0.01)$ | $1.00393_{\pm 0.01541}(0.05)$ | $1.00186_{\pm 0.00738}(0.11)$ |
| EGN-FT | $1.00906_{\pm 0.02962}(0.41)$ | $1.00193_{\pm 0.00927}(1.64)$ | $1.00071_{\pm 0.00442}(3.28)$ |
| EGN-FT-Online | $1.00761_{\pm 0.02794}(0.42)$ | $1.00131_{\pm 0.00792}(1.65)$ | $1.00036_{\pm 0.00304}(3.29)$ |
| EGN-TACO | $1.00772_{\pm 0.02808}(0.41)$ | $1.00100_{\pm 0.00618}(1.66)$ | $1.00040_{\pm 0.00341}(3.32)$ |
| EGN-TACO-Online | $\mathbf{1.00206_{\pm 0.00942}(0.42)}$ | $\mathbf{1.00042_{\pm 0.00369}(1.65)}$ | $\mathbf{1.00017_{\pm 0.00216}(3.33)}$ |
| Meta-EGN | $1.00392_{\pm 0.01252}(0.01)$ | $1.00102_{\pm 0.00580}(0.05)$ | $1.00073_{\pm 0.00455}(0.11)$ |
| Meta-EGN-FT | $1.00214_{\pm 0.00826}(0.41)$ | $1.00062_{\pm 0.00446}(1.65)$ | $1.00053_{\pm 0.00425}(3.29)$ |
| Meta-EGN-FT-Online | $1.00173_{\pm 0.00819}(0.41)$ | $1.00052_{\pm 0.00429}(1.64)$ | $1.00031_{\pm 0.00337}(3.29)$ |
| Meta-EGN-TACO | $\mathbf{1.00148_{\pm 0.00641}(0.41)}$ | $1.00035_{\pm 0.00304}(1.66)$ | $\mathbf{1.00027_{\pm 0.00268}(3.31)}$ |
| Meta-EGN-TACO-Online | $1.00477_{\pm 0.03821}(0.42)$ | $1.00425_{\pm 0.04920}(1.66)$ | $1.00602_{\pm 0.05284}(3.32)$ |

| RB200 | | | |
|---|---|---|---|
| Method | Fast (1) | Balanced (4) | Accurate (8) |
| EGN | $1.03982_{\pm 0.01087}(0.02)$ | $1.03175_{\pm 0.00561}(0.09)$ | $1.02940_{\pm 0.00483}(0.18)$ |
| EGN-FT | $1.02878_{\pm 0.00565}(0.69)$ | $1.02409_{\pm 0.00428}(2.76)$ | $1.02243_{\pm 0.00495}(5.51)$ |
| EGN-FT-Online | $1.03074_{\pm 0.00648}(0.70)$ | $1.02395_{\pm 0.00454}(2.75)$ | $1.02125_{\pm 0.00473}(5.46)$ |
| EGN-TACO | $1.02758_{\pm 0.00530}(0.69)$ | $1.02214_{\pm 0.00462}(2.76)$ | $1.02052_{\pm 0.00420}(5.52)$ |
| EGN-TACO-Online | $\mathbf{1.02678_{\pm 0.00631}(0.70)}$ | $\mathbf{1.02168_{\pm 0.00469}(2.75)}$ | $1.01930_{\pm 0.00461}(5.55)$ |
| Meta-EGN | $1.03394_{\pm 0.00746}(0.02)$ | $1.02655_{\pm 0.00496}(0.09)$ | $1.02502_{\pm 0.00478}(0.18)$ |
| Meta-EGN-FT | $1.02622_{\pm 0.00564}(0.68)$ | $1.02152_{\pm 0.00429}(2.73)$ | $1.01994_{\pm 0.00429}(5.47)$ |
| Meta-EGN-FT-Online | $1.02778_{\pm 0.00745}(0.68)$ | $1.02121_{\pm 0.00587}(2.73)$ | $1.02009_{\pm 0.00465}(5.54)$ |
| Meta-EGN-TACO | $\mathbf{1.02614_{\pm 0.00535}(0.69)}$ | $1.02143_{\pm 0.00505}(2.75)$ | $1.01979_{\pm 0.00474}(5.49)$ |
| Meta-EGN-TACO-Online | $1.03018_{\pm 0.01154}(0.69)$ | $\mathbf{1.02030_{\pm 0.00551}(2.78)}$ | $\mathbf{1.01903_{\pm 0.00497}(5.51)}$ |

| RB500 | | | |
|---|---|---|---|
| Method | Fast (1) | Balanced (4) | Accurate (8) |
| EGN | $1.02837_{\pm 0.00638}(0.06)$ | $1.02396_{\pm 0.00207}(0.23)$ | $1.02308_{\pm 0.00190}(0.46)$ |
| EGN-FT | $1.01951_{\pm 0.00284}(1.79)$ | $1.01699_{\pm 0.00229}(7.15)$ | $1.01630_{\pm 0.00208}(14.30)$ |
| EGN-FT-Online | $1.01846_{\pm 0.00251}(1.77)$ | $1.01653_{\pm 0.00226}(7.10)$ | $1.01518_{\pm 0.00197}(14.24)$ |
| EGN-TACO | $1.01878_{\pm 0.00280}(1.78)$ | $1.01663_{\pm 0.00230}(7.12)$ | $1.01577_{\pm 0.00214}(14.24)$ |
| EGN-TACO-Online | $\mathbf{1.01749_{\pm 0.00258}(1.77)}$ | $\mathbf{1.01540_{\pm 0.00202}(7.12)}$ | $\mathbf{1.01512_{\pm 0.00209}(14.25)}$ |
| Meta-EGN | $1.02328_{\pm 0.00302}(0.06)$ | $1.02065_{\pm 0.00226}(0.23)$ | $1.01983_{\pm 0.00229}(0.46)$ |
| Meta-EGN-FT | $1.01976_{\pm 0.00272}(1.78)$ | $1.01778_{\pm 0.00213}(7.10)$ | $1.01698_{\pm 0.00214}(14.21)$ |
| Meta-EGN-FT-Online | $1.01866_{\pm 0.00315}(1.78)$ | $1.01591_{\pm 0.00256}(7.07)$ | $1.01478_{\pm 0.00222}(14.20)$ |
| Meta-EGN-TACO | $1.01959_{\pm 0.00279}(1.78)$ | $1.01776_{\pm 0.00219}(7.11)$ | $1.01700_{\pm 0.00211}(14.23)$ |
| Meta-EGN-TACO-Online | $\mathbf{1.01566_{\pm 0.00360}(1.77)}$ | $\mathbf{1.01356_{\pm 0.00294}(7.08)}$ | $\mathbf{1.01213_{\pm 0.00273}(14.23)}$ |

nodes as the input, we examined the performance of the baselines and TACO with different numbers of random input initializations (seeds) in our experiments. Consistent with Karalias & Loukas (2020) and Wang & Li (2023), we take the seed leading to the best solution as the final output when using multiple input initializations.

## 4.3 EMPIRICAL RESULTS

**Static problems.** We begin by evaluating the compatibility of TACO with EGN and Meta-EGN. Figures 3 and 4 report the mean ApR as a function of the number of gradient update steps across all datasets and tasks, using EGN and Meta-EGN as backbones, respectively. All results are obtained by setting the number of seeds to 4 for all models. Compared to the baseline methods, where we update

Table 2: Mean ApR and seconds per graph of all methods on MC ($\uparrow$). All results are from 30 update steps. "FT" stands for fine-tuning; "Accurate (8)" means 8 seeds were used. The best solutions are in **bold**, and the cases where EGN + TACO outperform Meta-EGN-FT(-Online) are in gray.

| TWITTER | | | |
|---|---|---|---|
| Method | Fast (1) | Balanced (4) | Accurate (8) |
| EGN | $0.73856_{\pm 0.25477}(0.01)$ | $0.91233_{\pm 0.12556}(0.03)$ | $0.95073_{\pm 0.08131}(0.06)$ |
| EGN-FT | $0.93744_{\pm 0.13083}(0.23)$ | $0.98332_{\pm 0.06204}(0.91)$ | $0.99151_{\pm 0.04802}(1.83)$ |
| EGN-FT-Online | $0.94610_{\pm 0.12316}(0.24)$ | $0.98182_{\pm 0.06225}(0.94)$ | $0.98672_{\pm 0.04891}(1.84)$ |
| EGN-TACO | $\mathbf{0.95419}_{\pm \mathbf{0.10578}}(\mathbf{0.23})$ | $\mathbf{0.99295}_{\pm \mathbf{0.02656}}(\mathbf{0.93})$ | $\mathbf{0.99766}_{\pm \mathbf{0.01310}}(\mathbf{1.86})$ |
| EGN-TACO-Online | $0.95119_{\pm 0.10600}(0.24)$ | $0.98400_{\pm 0.05507}(0.95)$ | $0.99277_{\pm 0.03325}(1.90)$ |
| Meta-EGN | $0.91078_{\pm 0.12812}(0.01)$ | $0.95158_{\pm 0.09174}(0.03)$ | $0.96538_{\pm 0.07509}(0.06)$ |
| Meta-EGN-FT | $\mathbf{0.94930}_{\pm \mathbf{0.09735}}(\mathbf{0.24})$ | $0.97601_{\pm 0.06939}(0.94)$ | $0.98749_{\pm 0.05144}(1.89)$ |
| Meta-EGN-FT-Online | $0.94836_{\pm 0.09920}(0.24)$ | $0.97430_{\pm 0.06951}(0.94)$ | $0.98253_{\pm 0.05748}(1.85)$ |
| Meta-EGN-TACO | $0.94783_{\pm 0.10246}(0.24)$ | $0.97799_{\pm 0.06808}(0.95)$ | $0.98663_{\pm 0.05393}(1.89)$ |
| Meta-EGN-TACO-Online | $0.94625_{\pm 0.11002}(0.24)$ | $\mathbf{0.98244}_{\pm \mathbf{0.05885}}(\mathbf{0.94})$ | $\mathbf{0.99061}_{\pm \mathbf{0.03641}}(\mathbf{1.89})$ |
| COLLAB | | | |
| Method | Fast (1) | Balanced (4) | Accurate (8) |
| EGN | $0.84956_{\pm 0.29397}(0.01)$ | $0.97625_{\pm 0.10808}(0.04)$ | $0.99663_{\pm 0.02481}(0.07)$ |
| EGN-FT | $0.90862_{\pm 0.22514}(0.29)$ | $0.98747_{\pm 0.07870}(1.15)$ | $0.99916_{\pm 0.01238}(2.30)$ |
| EGN-FT-Online | $0.98019_{\pm 0.09857}(0.29)$ | $0.99930_{\pm 0.01114}(1.19)$ | $0.99969_{\pm 0.00693}(2.37)$ |
| EGN-TACO | $0.95378_{\pm 0.15099}(0.30)$ | $0.99631_{\pm 0.03368}(1.18)$ | $0.99976_{\pm 0.00580}(2.36)$ |
| EGN-TACO-Online | $\mathbf{0.98217}_{\pm \mathbf{0.09103}}(\mathbf{0.30})$ | $\mathbf{0.99937}_{\pm \mathbf{0.00931}}(\mathbf{1.20})$ | $\mathbf{1.00000}_{\pm \mathbf{0.00000}}(\mathbf{2.40})$ |
| Meta-EGN | $0.98965_{\pm 0.05931}(0.01)$ | $0.99721_{\pm 0.02583}(0.04)$ | $0.99813_{\pm 0.01932}(0.08)$ |
| Meta-EGN-FT | $0.99546_{\pm 0.02957}(0.29)$ | $0.99912_{\pm 0.01169}(1.18)$ | $0.99966_{\pm 0.00647}(2.35)$ |
| Meta-EGN-FT-Online | $0.99381_{\pm 0.03670}(0.29)$ | $0.99894_{\pm 0.01298}(1.19)$ | $0.99961_{\pm 0.00735}(2.37)$ |
| Meta-EGN-TACO | $0.99526_{\pm 0.03094}(0.30)$ | $0.99912_{\pm 0.01169}(1.19)$ | $0.99980_{\pm 0.00464}(2.38)$ |
| Meta-EGN-TACO-Online | $\mathbf{0.99787}_{\pm \mathbf{0.01923}}(\mathbf{0.30})$ | $\mathbf{0.99978}_{\pm \mathbf{0.00516}}(\mathbf{1.19})$ | $\mathbf{0.99990}_{\pm \mathbf{0.00301}}(\mathbf{2.41})$ |
| RB200 | | | |
| Method | Fast (1) | Balanced (4) | Accurate (8) |
| EGN | $0.76354_{\pm 0.14769}(0.01)$ | $0.84750_{\pm 0.14285}(0.03)$ | $0.91045_{\pm 0.11979}(0.06)$ |
| EGN-FT | $0.80304_{\pm 0.15676}(0.23)$ | $0.90088_{\pm 0.13542}(0.91)$ | $0.95659_{\pm 0.09698}(1.83)$ |
| EGN-FT-Online | $0.88725_{\pm 0.14130}(0.24)$ | $0.96475_{\pm 0.07824}(0.95)$ | $\mathbf{0.99455}_{\pm \mathbf{0.02159}}(\mathbf{1.90})$ |
| EGN-TACO | $0.84914_{\pm 0.15928}(0.23)$ | $0.94480_{\pm 0.10703}(0.94)$ | $0.98094_{\pm 0.05496}(1.88)$ |
| EGN-TACO-Online | $\mathbf{0.90714}_{\pm \mathbf{0.12065}}(\mathbf{0.24})$ | $\mathbf{0.97687}_{\pm \mathbf{0.06840}}(\mathbf{0.98})$ | $0.98465_{\pm 0.05296}(1.88)$ |
| Meta-EGN | $0.77893_{\pm 0.18384}(0.01)$ | $0.87345_{\pm 0.13786}(0.03)$ | $0.89779_{\pm 0.11543}(0.06)$ |
| Meta-EGN-FT | $0.87840_{\pm 0.14259}(0.24)$ | $0.93146_{\pm 0.10693}(0.98)$ | $0.94966_{\pm 0.09027}(1.95)$ |
| Meta-EGN-FT-Online | $0.90159_{\pm 0.12417}(0.25)$ | $0.93530_{\pm 0.09976}(0.98)$ | $0.95546_{\pm 0.07421}(1.98)$ |
| Meta-EGN-TACO | $0.88296_{\pm 0.14071}(0.25)$ | $0.93723_{\pm 0.10152}(1.00)$ | $0.94984_{\pm 0.09018}(1.99)$ |
| Meta-EGN-TACO-Online | $\mathbf{0.91264}_{\pm \mathbf{0.11546}}(\mathbf{0.25})$ | $\mathbf{0.96069}_{\pm \mathbf{0.07413}}(\mathbf{1.01})$ | $\mathbf{0.97881}_{\pm \mathbf{0.05873}}(\mathbf{1.93})$ |
| RB500 | | | |
| Method | Fast (1) | Balanced (4) | Accurate (8) |
| EGN | $0.80616_{\pm 0.20549}(0.01)$ | $0.83771_{\pm 0.19788}(0.06)$ | $0.88334_{\pm 0.17438}(0.12)$ |
| EGN-FT | $0.82545_{\pm 0.19791}(0.46)$ | $0.90312_{\pm 0.16580}(1.84)$ | $0.95306_{\pm 0.11117}(3.68)$ |
| EGN-FT-Online | $\mathbf{0.87644}_{\pm \mathbf{0.18240}}(\mathbf{0.49})$ | $0.93703_{\pm 0.14093}(1.95)$ | $0.98121_{\pm 0.06640}(3.82)$ |
| EGN-TACO | $0.83504_{\pm 0.19576}(0.47)$ | $0.92717_{\pm 0.14329}(1.88)$ | $0.97101_{\pm 0.07950}(3.75)$ |
| EGN-TACO-Online | $0.86607_{\pm 0.18805}(0.48)$ | $\mathbf{0.94717}_{\pm \mathbf{0.13002}}(\mathbf{1.98})$ | $\mathbf{0.98684}_{\pm \mathbf{0.04781}}(\mathbf{3.88})$ |
| Meta-EGN | $0.79767_{\pm 0.20632}(0.01)$ | $0.84136_{\pm 0.19527}(0.06)$ | $0.88511_{\pm 0.17458}(0.12)$ |
| Meta-EGN-FT | $0.80896_{\pm 0.20270}(0.45)$ | $0.86025_{\pm 0.18965}(1.80)$ | $0.91018_{\pm 0.15735}(3.61)$ |
| Meta-EGN-FT-Online | $0.82832_{\pm 0.19989}(0.47)$ | $0.90237_{\pm 0.17033}(1.89)$ | $0.95514_{\pm 0.11758}(3.75)$ |
| Meta-EGN-TACO | $0.82316_{\pm 0.19892}(0.45)$ | $0.87036_{\pm 0.18832}(1.81)$ | $0.91610_{\pm 0.15453}(3.63)$ |
| Meta-EGN-TACO-Online | $\mathbf{0.86786}_{\pm \mathbf{0.18423}}(\mathbf{0.48})$ | $\mathbf{0.91854}_{\pm \mathbf{0.15807}}(\mathbf{1.89})$ | $\mathbf{0.96691}_{\pm \mathbf{0.09560}}(\mathbf{3.78})$ |

the parameters of a trained model or a freshly initialized model, TACO consistently achieves superior performance across a wide range of update budgets. More importantly, within 10 update steps, TACO can achieve solutions unattainable by naively fine-tuning trained models for 30 steps, and TACO can outperform optimizing freshly initialized models when fine-tuning trained models falls short. As mentioned earlier, even though Meta-EGN enables instance-wise adaptability, its adaptability can still be improved when paired with TACO.

Next, we assess the robustness of all methods under varying numbers of random seeds. Tables 1 and 2 summarize these results. Models enhanced with TACO consistently achieve the best performance across almost all settings, with the online version potentially offering additional gains. We also include the mean ApR of EGN with 256 seeds in Table 9 in Appendix B. In nearly all cases, TACO-

Table 3: Mean ApR and seconds per graph of all methods under distribution shift.

| Method | MVC ($\downarrow$) | MC ($\uparrow$) |
|---|---|---|
| EGN | $1.05976_{\pm0.00737}(0.18)$ | $0.90586_{\pm0.11876}(0.06)$ |
| EGN-FT | $1.05453_{\pm0.00692}(5.50)$ | $0.98558_{\pm0.03514}(1.92)$ |
| EGN-FT-Online | $1.02505_{\pm0.01135}(5.53)$ | $0.98148_{\pm0.06051}(1.91)$ |
| EGN-TACO | $1.03659_{\pm0.00617}(5.50)$ | $\mathbf{0.99166_{\pm0.03770}(1.92)}$ |
| EGN-TACO-Online | $\mathbf{1.01958_{\pm0.00454}(5.56)}$ | $0.98703_{\pm0.04886}(1.95)$ |
| Meta-EGN | $1.04744_{\pm0.00702}(0.18)$ | $0.90362_{\pm0.10715}(0.06)$ |
| Meta-EGN-FT | $1.03044_{\pm0.00591}(5.52)$ | $0.93951_{\pm0.09678}(1.97)$ |
| Meta-EGN-FT-Online | $1.02354_{\pm0.00553}(5.55)$ | $0.97317_{\pm0.05384}(1.94)$ |
| Meta-EGN-TACO | $1.02875_{\pm0.00533}(5.52)$ | $0.94491_{\pm0.09504}(1.97)$ |
| Meta-EGN-TACO-Online | $\mathbf{1.01975_{\pm0.00558}(5.56)}$ | $\mathbf{0.97328_{\pm0.05842}(1.97)}$ |

Table 4: Mean ApR and seconds per graph of all methods on dynamic problems.

| Method | MVC ($\downarrow$) | MC ($\uparrow$) |
|---|---|---|
| EGN | $1.04315_{\pm0.06230}(0.14)$ | $0.82964_{\pm0.09868}(0.05)$ |
| EGN-FT | $1.01515_{\pm0.04569}(4.33)$ | $0.95712_{\pm0.08046}(1.42)$ |
| EGN-FT-Online | $1.01158_{\pm0.03530}(4.35)$ | $1.00000_{\pm0.00000}(1.38)$ |
| EGN-TACO | $1.01050_{\pm0.03281}(4.33)$ | $0.98402_{\pm0.05179}(1.42)$ |
| EGN-TACO-Online | $\mathbf{1.00852_{\pm0.02950}(4.30)}$ | $\mathbf{1.00000_{\pm0.00000}(1.38)}$ |
| Meta-EGN | $1.01244_{\pm0.03563}(0.14)$ | $0.82533_{\pm0.10378}(0.05)$ |
| Meta-EGN-FT | $0.99961_{\pm0.01819}(4.36)$ | $0.98476_{\pm0.05198}(1.59)$ |
| Meta-EGN-FT-Online | $1.01961_{\pm0.04904}(4.34)$ | $0.99353_{\pm0.03314}(1.46)$ |
| Meta-EGN-TACO | $\mathbf{0.99639_{\pm0.01366}(4.36)}$ | $0.99015_{\pm0.04546}(1.59)$ |
| Meta-EGN-TACO-Online | $1.00947_{\pm0.03128}(4.34)$ | $\mathbf{0.99413_{\pm0.03087}(1.38)}$ |

enhanced models discover better solutions in a comparable amount of wall-clock time, except for the MC task on RB200 and RB500. These exceptions may be attributed to the nature of RB200 and RB500. Since the cliques are generated deliberately, and the random one-hot input vector can be interpreted as an initial guess, in the extreme setting, exhaustive exploration of initial guesses leads to strong performance. Nevertheless, we note that our goal is not to beat EGN and Meta-EGN with a large number of runs in comparable or less runtime and many fewer runs. TACO is orthogonal to the number of seeds, and the different runs of EGN and Meta-EGN can be executed in parallel, with each run paired with TACO, so the runtime does not scale linearly with the number of runs.

Although TACO is a model-agnostic framework to enhance unsupervised NCO models and not explicitly designed to compete with MAML in NCO, EGN with TACO can surpass fine-tuned Meta-EGN in about half of the cases in the MVC experiments and nearly all cases in the MC experiments, as highlighted in Tables 1 and 2.

**Distribution shift.** The detailed results are presented in Table 3. All models were tuned with 30 update steps, and 8 seeds were used. Models enhanced with TACO consistently demonstrate greater robustness to shift, maintaining better ApR than the fine-tuned counterparts. The EGN models without any additional optimization can only achieve 1.05976 on MVC and 0.90586 on MC, whereas the EGN models trained on RB200 and tested on RB200 achieve 1.02940 on MVC and 0.91045 on MC. For Meta-EGN, a similar performance drop on MVC can be observed (1.04744 vs. 1.02502), but it remains robust on MC with distribution shift (0.90362 vs. 0.89779), which aligns with the findings of Wang & Li (2023).

**Dynamic problems.** As detailed in Table 4, models enhanced with TACO achieve superior performance on both the dynamic MVC and dynamic MC problems, maintaining the highest mean ApRs. These results highlight TACO's effectiveness in guiding models toward high-quality solutions in evolving environments, thereby broadening its applicability to dynamic problem settings. Ideally, the online version is expected to work better than the standard version for dynamic problems, but this largely depends on the degree of problem-specific structural change in the graph snapshots over time: when there is little structural overlap, the parameters from the previous snapshot would be less useful (Liao et al., 2025).

**Sensitivity analysis on SP parameters.** To validate the generality and robustness of TACO, we selected the SP parameters relatively uniformly across datasets with limited tuning. This ensures that the observed performance gains are not the result of dataset-specific overfitting, but instead stem from the effectiveness of TACO. We include results of TACO with different sets of SP parameters in

Table 10 in Appendix B. The results demonstrate that TACO consistently outperforms baselines across different parameter choices, suggesting that TACO is not overly sensitive to hyperparameter settings.

## 5 RELATED WORK

The supervised learning paradigm has been shown to be powerful in NCO (Joshi et al., 2019; 2022; Vinyals et al., 2015; Gasse et al., 2019; Sun & Yang, 2023; Hudson et al., 2022; Li et al., 2023; 2024). These methods train models to predict high-quality solutions by leveraging large datasets of problem instances annotated with optimal or near-optimal solutions. However, producing such labels is computationally expensive, particularly for large-scale instances.

UL and reinforcement learning (RL) approaches have been proposed to mitigate this dependency on labeled data (Bello et al., 2016; Khalil et al., 2017; Kool et al., 2019; Karalias & Loukas, 2020; Qiu et al., 2022; Toenshoff et al., 2021; Tönshoff et al., 2023; Wang & Li, 2023; Sanokowski et al., 2023). Despite this advantage, most UL and RL-based approaches still rely on extensive offline training across large datasets to learn *heuristics that generalize* across instances. An alternative instance-specific paradigm was introduced by Schuetz et al. (2022), who proposed an unsupervised framework that learns *instance-specific heuristics* by directly optimizing the combinatorial objective on a per-instance basis. This approach bypasses the need for offline training entirely, enabling the model to adapt to individual problem instances at test time. Follow-up works have enhanced this framework by improving solution quality, incorporating higher-order reasoning, and addressing dynamic CO problems (Heydaribeni et al., 2024; Ichikawa, 2024; Liao et al., 2025), achieving robust performance even on large-scale graphs.

Our approach is also related to Test-Time Training (Sun et al., 2020), which enhances supervised models during inference by optimizing on an auxiliary self-supervised task. However, in UL-based NCO, where models are trained using an unsupervised problem-specific objective, an auxiliary task is not needed. Instead, the UL objective used in training can be reused to guide test-time adaptation. More broadly, our method falls under the umbrella of the Test-Time Adaptation paradigm (Liang et al., 2025), which seeks to adapt trained models at test-time. In the NCO domain, prior works have primarily focused on improving solution quality during inference for RL-based approaches. Hottung et al. (2022) developed Efficient Active Search that updates a subset of model parameters for each test instance. Meta-SAGE (Son et al., 2023) adapts the model at test-time for better scalability. COMPASS (Chalumeau et al., 2023) employs search in a latent space to enable instance-specific policy adaptation. Our work extends the frontier of test-time adaptation to unsupervised NCO. In contrast to Meta-EGN, we accomplish effective adaptation through the lens of principled warm-starting and simultaneously unify generalizable and instance-specific NCO.

## 6 DISCUSSION

**Conclusion.** We introduced TACO, a model-agnostic test-time adaptation framework that bridges the gap between generalizable and instance-specific NCO. By viewing instance-wise adaptation from a warm-starting perspective, TACO combines the strengths of both paradigms, leveraging learned hypotheses while enabling effective instance-level refinement. Our extensive experiments on classical problems, Minimum Vertex Cover and Maximum Clique, demonstrate that TACO consistently improves solution quality across static, distribution-shifted, and dynamic settings, all while incurring negligible computational overhead compared to standard fine-tuning. These results highlight the broad applicability and practical benefits of integrating TACO into unsupervised NCO pipelines.

**Limitations and future work.** The reported runtimes could be significantly reduced by enhancing the backbones, EGN and Meta-EGN, through parallelization of the seed dimension, adoption of more sophisticated input feature designs, and more efficient decoding mechanisms. Additionally, if batch data is available at test time, curriculum learning (Bengio et al., 2009; Lisicki et al., 2020; Liu et al., 2024) could be incorporated into TACO. Exploring training strategies that explicitly encourage compatibility with TACO could also potentially accelerate convergence and enable fast transfer across related CO problems.

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

# A  ADDITIONAL IMPLEMENTATION DETAILS

We provide the exact loss function used in our experiments. For detailed derivations, please refer to Karalias & Loukas (2020) and Wang & Li (2023). The MVC loss is defined as:

$$\ell_{\mathrm{MVC}}(D; G) = \sum_{i=1}^{n} x_i + \beta \sum_{(i,j) \in E} (1 - x_i)(1 - x_j).$$

We adopted the simplified MC loss same as the implementation by Karalias & Loukas (2020):

$$\ell_{\mathrm{MC}}(D; G) = -\frac{1}{2} \sum_{(i,j) \in E} x_i x_j + \frac{\beta}{2} \sum_{i \neq j} x_i x_j.$$

We followed the training settings described in Wang & Li (2023). $\beta$ was set to 0.5 for the MVC experiments and 4 for the MC experiments. For tuning the trained models, we set $\beta = 0.5$ and $\lambda_{\mathrm{perturb}} = 0.001$ in all experiments; we used 0.0001 as the learning rate for tuning EGN models for MVC, 0.001 for MC, and 0.001 for Meta-EGN models for both problems.

**Hyperparameters for static problems.** The shrink parameter used in all experiments is listed in Tables 5 and 6.

Table 5: $\lambda_{\mathrm{shrink}}$ used in experiments for static MVC problems.

| | Twitter | | COLLAB | | RB200 | | RB500 | |
|---|---|---|---|---|---|---|---|---|
| Method | $\lambda_{\mathrm{shrink}}$ | $\lambda_{\mathrm{shrink\text{-}online}}$ | $\lambda_{\mathrm{shrink}}$ | $\lambda_{\mathrm{shrink\text{-}online}}$ | $\lambda_{\mathrm{shrink}}$ | $\lambda_{\mathrm{shrink\text{-}online}}$ | $\lambda_{\mathrm{shrink}}$ | $\lambda_{\mathrm{shrink\text{-}online}}$ |
| EGN-TACO | 0.3 | - | 0.3 | - | 0.3 | - | 0.5 | - |
| EGN-TACO-Online | 0.3 | 0.99 | 0.3 | 0.99 | 0.3 | 0.99 | 0.5 | 0.99 |
| Meta-EGN-TACO | 0.7 | - | 0.7 | - | 0.7 | - | 0.9 | - |
| Meta-EGN-TACO-Online | 0.7 | 0.9 | 0.7 | 0.9 | 0.7 | 0.9 | 0.9 | 0.9 |

Table 6: $\lambda_{\mathrm{shrink}}$ used in experiments for static MC problems.

| | Twitter | | COLLAB | | RB200 | | RB500 | |
|---|---|---|---|---|---|---|---|---|
| Method | $\lambda_{\mathrm{shrink}}$ | $\lambda_{\mathrm{shrink\text{-}online}}$ | $\lambda_{\mathrm{shrink}}$ | $\lambda_{\mathrm{shrink\text{-}online}}$ | $\lambda_{\mathrm{shrink}}$ | $\lambda_{\mathrm{shrink\text{-}online}}$ | $\lambda_{\mathrm{shrink}}$ | $\lambda_{\mathrm{shrink\text{-}online}}$ |
| EGN-TACO | 0.3 | - | 0.3 | - | 0.3 | - | 0.5 | - |
| EGN-TACO-Online | 0.3 | 0.99 | 0.3 | 0.99 | 0.3 | 0.99 | 0.5 | 0.99 |
| Meta-EGN-TACO | 0.7 | - | 0.7 | - | 0.7 | - | 0.7 | - |
| Meta-EGN-TACO-Online | 0.7 | 0.9 | 0.7 | 0.9 | 0.7 | 0.9 | 0.7 | 0.99 |

**Hyperparameters for problems with distribution shift.** The shrink parameter used in all experiments is listed in Tables 7.

Table 7: $\lambda_{\mathrm{shrink}}$ used in experiments for distribution shift.

| | MVC | | MC | |
|---|---|---|---|---|
| Method | $\lambda_{\mathrm{shrink}}$ | $\lambda_{\mathrm{shrink\text{-}online}}$ | $\lambda_{\mathrm{shrink}}$ | $\lambda_{\mathrm{shrink\text{-}online}}$ |
| EGN-TACO | 0.3 | - | 0.3 | - |
| EGN-TACO-Online | 0.3 | 0.99 | 0.3 | 0.99 |
| Meta-EGN-TACO | 0.7 | - | 0.7 | - |
| Meta-EGN-TACO-Online | 0.7 | 0.9 | 0.7 | 0.9 |

**Hyperparameters for dynamic problems.** The shrink parameter used in all experiments is listed in Tables 8.

Table 8: $\lambda_{\text{shrink}}$ used in experiments for dynamic problems.

| Method | MVC | | MC | |
| --- | --- | --- | --- | --- |
| | $\lambda_{\text{shrink}}$ | $\lambda_{\text{shrink-online}}$ | $\lambda_{\text{shrink}}$ | $\lambda_{\text{shrink-online}}$ |
| EGN-TACO | 0.5 | - | 0.5 | - |
| EGN-TACO-Online | 0.5 | 1 | 0.5 | 1 |
| Meta-EGN-TACO | 0.5 | - | 0.5 | - |
| Meta-EGN-TACO-Online | 0.5 | 1 | 0.5 | 1 |

All models were implemented using PyTorch (Paszke et al., 2019) and PyTorch Geometric (Fey & Lenssen, 2019). Experiments were conducted on a machine with a single NVIDIA GeForce RTX 4090 GPU, a 32-core Intel Core i9-14900K CPU, and 64 GB of RAM running Ubuntu 24.04.

# B    ADDITIONAL RESULTS

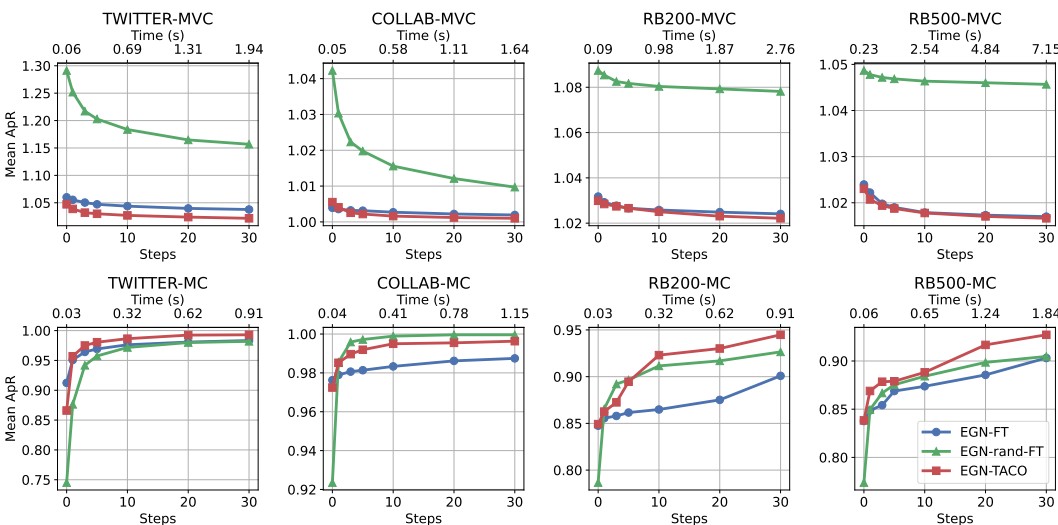

Figure 5: Mean ApR of methods using EGN as the backbone on static MVC ($\downarrow$) and MC ($\uparrow$) problems with respect to the number of update steps. "FT" stands for fine-tuning; "rand" means models are freshly initialized. The wall clock time factors in the decoding operations.

Table 9: Mean ApR and seconds per graph of EGN with 256 seeds and TACO-enhanced EGN with 8 seeds.

| TWITTER | | |
|---|---|---|
| Method | MVC ($\downarrow$) | MC ($\uparrow$) |
| EGN (256) | $1.02875_{\pm 0.02073}(3.37)$ | $0.99167_{\pm 0.02903}(1.03)$ |
| EGN-TACO (8) | $1.01668_{\pm 0.01636}(3.90)$ | $0.99766_{\pm 0.01310}(1.86)$ |
| EGN-TACO-Online (8) | $1.01357_{\pm 0.01417}(3.96)$ | $0.99277_{\pm 0.03325}(1.90)$ |
| COLLAB | | |
| Method | MVC ($\downarrow$) | MC ($\uparrow$) |
| EGN (256) | $1.00040_{\pm 0.00319}(2.70)$ | $1.00000_{\pm 0.00000}(1.68)$ |
| EGN-TACO (8) | $1.00040_{\pm 0.00341}(3.32)$ | $0.99976_{\pm 0.00580}(2.36)$ |
| EGN-TACO-Online (8) | $1.00017_{\pm 0.00216}(3.33)$ | $1.00000_{\pm 0.00000}(2.40)$ |
| RB200 | | |
| Method | MVC ($\downarrow$) | MC ($\uparrow$) |
| EGN (256) | $1.02071_{\pm 0.00405}(5.01)$ | $0.99575_{\pm 0.01858}(1.07)$ |
| EGN-TACO (8) | $1.02052_{\pm 0.00420}(5.52)$ | $0.98094_{\pm 0.05496}(1.88)$ |
| EGN-TACO-Online (8) | $1.01930_{\pm 0.00461}(5.55)$ | $0.98465_{\pm 0.05296}(1.88)$ |
| RB500 | | |
| Method | MVC ($\downarrow$) | MC ($\uparrow$) |
| EGN (256) | $1.01932_{\pm 0.00177}(13.77)$ | $0.99635_{\pm 0.01497}(2.44)$ |
| EGN-TACO (8) | $1.01577_{\pm 0.00214}(14.24)$ | $0.97101_{\pm 0.07950}(3.75)$ |
| EGN-TACO-Online (8) | $1.01512_{\pm 0.00209}(14.25)$ | $0.98684_{\pm 0.04781}(3.88)$ |

Table 10: Mean ApR of EGN-TACO with different sets of SP parameters. $\lambda_{\text{shrink}} = 1, \lambda_{\text{perturb}} = 0$ is equivalent to EGN-FT. All settings use 30 update steps and 8 random seeds.

| $\lambda_{\text{shrink}}$ | $\lambda_{\text{perturb}}$ | TWITTER-MVC ($\downarrow$) | TWITTER-MC ($\uparrow$) | COLLAB-MVC ($\downarrow$) | COLLAB-MC ($\uparrow$) |
|---|---|---|---|---|---|
| 0.0 | 0.001 | $1.05593_{\pm 0.06882}$ | $0.67702_{\pm 0.24641}$ | $1.01325_{\pm 0.03691}$ | $0.84874_{\pm 0.28305}$ |
| 0.1 | 0.001 | $1.01233_{\pm 0.01254}$ | $0.99678_{\pm 0.01445}$ | $1.00023_{\pm 0.00243}$ | $0.99983_{\pm 0.00527}$ |
| 0.3 | 0.001 | $1.01668_{\pm 0.01636}$ | $0.99766_{\pm 0.01310}$ | $1.00040_{\pm 0.00341}$ | $0.99976_{\pm 0.00580}$ |
| 0.5 | 0.001 | $1.02221_{\pm 0.01871}$ | $0.99507_{\pm 0.03153}$ | $1.00039_{\pm 0.00342}$ | $0.99954_{\pm 0.00900}$ |
| 0.7 | 0.001 | $1.02563_{\pm 0.01986}$ | $0.99256_{\pm 0.03468}$ | $1.00043_{\pm 0.00336}$ | $0.99954_{\pm 0.00900}$ |
| 0.9 | 0.001 | $1.02845_{\pm 0.02093}$ | $0.99193_{\pm 0.04797}$ | $1.00048_{\pm 0.00369}$ | $0.99916_{\pm 0.01238}$ |
| 0.3 | 0.0001 | $1.01690_{\pm 0.01646}$ | $0.99781_{\pm 0.01355}$ | $1.00039_{\pm 0.00327}$ | $0.99956_{\pm 0.00857}$ |
| 0.3 | 0.001 | $1.01668_{\pm 0.01636}$ | $0.99766_{\pm 0.01310}$ | $1.00040_{\pm 0.00341}$ | $0.99976_{\pm 0.00580}$ |
| 0.3 | 0.01 | $1.01646_{\pm 0.01568}$ | $0.99775_{\pm 0.01056}$ | $1.00032_{\pm 0.00299}$ | $0.99954_{\pm 0.00900}$ |
| 0.3 | 0.1 | $1.02063_{\pm 0.01765}$ | $0.99827_{\pm 0.01340}$ | $1.00028_{\pm 0.00265}$ | $0.99929_{\pm 0.01197}$ |
| 1.0 | 0 | $1.03026_{\pm 0.02123}$ | $0.99151_{\pm 0.04802}$ | $1.00071_{\pm 0.00442}$ | $0.99916_{\pm 0.01238}$ |

# C   USE OF LARGE LANGUAGE MODELS

Large language models were used for editing purposes only.

