# OpenReview forum: "Test-Time Adaptation for Unsupervised Combinatorial Optimization"
_ICLR.cc/2026/Conference — ICLR 2026 Conference Withdrawn Submission_

### Official Review · Reviewer_q5YS · 2025-10-19

**Soundness:** 3
**Presentation:** 2
**Contribution:** 1
**Rating:** 2
**Confidence:** 4

**Summary:**

This paper introduces TACO (Test-time Adaptation for unsupervised Combinatorial Optimization), a model-agnostic framework that unifies two dominant paradigms in unsupervised combinatorial optimization (UCO): generalization-focused (pre-trained models for unseen instances, e.g., EGN, Meta-EGN) and instance-specific (optimized from scratch per instance, e.g., PI-GNN). TACO leverages a shrink-and-perturb (SP) warm-start strategy to adapt pre-trained models to individual test instances. Experiemnts are conducted on Minimum Vertex Cover (MVC) and Maximum Clique (MC) problems.

**Strengths:**

1. The authors identified a critical gap in NCO practices: how to unify generalization-focused and instance-specific paradigms for unsupervised learning for CO.
2. The proposed shrink-and-perturb (SP) technique for initializing the test-time model adaption is interesting, and its plug-and-play manner could potentially synergize with existing UCO pipelines.

**Weaknesses:**

1. The overall contribution is somewhat limited. The proposed SP trick, as the main technical contribution in this work, is quite straightforward and intuitive, which seems to be simply motivated empirically (Fig. 1) and by citation to (Ash & Adams, 2020) in supervised learning.
    - Could you give some theoretical analysis or explanation regarding why shrinking $\lambda_{\text{shrink}}\theta$ toward zero and adding Gaussian noise escapes the poor local minima in which vanilla fine-tuning gets stuck?
    - Or more specifically, the statement around line 60 that "*this is because the optimization landscape around trained parameters may be less conducive to rapid adaptation, potentially due to overfitting or local minima*" seems to lack grounded proof or more essential illustrations beyond the current result-oriented speculations.

2. Most importantly, the scope of experimental evaluation is quite insufficient.
    - TACO is only evaluated on MVC and Max Clique, which are among the simplest CO problems on graphs. So, it remains unproven for edge-focused or routing problems (e.g., TSP, VRP), scheduling tasks, etc., especially those with complex constraints which are core to NCO. I understand that some tasks can hardly be formulated and learned via unsupervised learning, then I suggest the authors provide empirical results on at least more node-focused CO problems (if not broader) like Maximum Independent Set (MIS), Maximum Cut, etc. Currently, TACO's claimed "broad applicability" and the title's reference to "combinatorial optimization" (a broad field encompassing non-graph, constrained, and sequential tasks) are weakened.
    - The paper claims its method to be model-agnostic, but all the evaluations are conducted solely upon (Meta-)EGN. Results that demonstrate TACO's adaptability to other solvers are lacking.
    - The comparisons are quite incomplete with a wide range of neural baselines missing. The authors only compare TACO with (Meta-)EGN and its ablation variants. I recommend the authors compare their method with (**at least a substantial subset of**) more powerful neural baselines employing diverse learning paradigms, e.g., DiffUCO[1], RLSA[2], iSCO[3], COExpander[4], VAG-CO[5], GFlowNets[6], and many more works that have incorporated more types of CO problems (e.g., [7-13], etc.). These listed methods have shown very competitive performance on the node-selection tasks and are conventionally compared by methods published at similar top conferences.
    - Minor advice: could you please report the results in a more recent and standard fashion in the NCO community with 1) objective value (e.g., the absolute size for the selected node set) including this metric computed by an optimal baseline (e.g., Gurobi), 2) relative optimality gap (e.g., |ApR - 1| * 100%), and 3) per-instance solving time. This is for a more transparent and comparable evaluation against existing and future works.


### References:
1) Methods focused on node-selection CO tasks:

[1] A Diffusion Model Framework for Unsupervised Neural Combinatorial Optimization

[2] Regularized Langevin Dynamics for Combinatorial Optimization

[3] Revisiting Sampling for Combinatorial Optimization

[4] COExpander: Adaptive Solution Expansion for Combinatorial Optimization

[5] Variational Annealing on Graphs for Combinatorial Optimization

[6] Let the Flows Tell: Solving Graph Combinatorial Optimization Problems with GFlowNets

2) Beyond:

[7] Sym-NCO: Leveraging Symmetricity for Neural Combinatorial Optimization

[8] DIMES: A Differentiable Meta Solver for Combinatorial Optimization Problems

[9] Unsupervised Learning for Solving the Travelling Salesman Problem

[10] DIFUSCO: Graph-based Diffusion Solvers for Combinatorial Optimization

[11] Learning What to Defer for Maximum Independent Sets

[12] Fast T2T: Optimization Consistency Speeds Up Diffusion-Based Training-to-Testing Solving for Combinatorial Optimization

[13] UniCO: On Unified Combinatorial Optimization via Problem Reduction to Matrix-Encoded General TSP

**Questions:**

For convenience, please refer to the Weaknesses part where I have listed my main concerns.

---

### Official Review · Reviewer_9Hcs · 2025-10-27

**Soundness:** 3
**Presentation:** 2
**Contribution:** 1
**Rating:** 4
**Confidence:** 4

**Summary:**

This paper proposes TACO (Test-time Adaptation for unsupervised Combinatorial Optimization), a model-agnostic framework that bridges the gap between generalization-focused and instance-specific paradigms in unsupervised neural combinatorial optimization (NCO). The key insight is that naive fine-tuning of pre-trained models often underperforms compared to training from scratch on individual instances. TACO addresses this by using a "shrink and perturb" (SP) warm-starting technique that modifies pre-trained parameters before test-time adaptation: $\theta^* \leftarrow \lambda_{shrink} \cdot \theta + \lambda_{perturb} \cdot \epsilon$, where $\epsilon \sim \mathcal{N}(0, \sigma^2)$. This approach preserves learned inductive biases while enabling more effective exploration during adaptation. The method is evaluated on Minimum Vertex Cover and Maximum Clique problems across static, distribution-shifted, and dynamic settings, demonstrating consistent improvements over baselines with negligible computational overhead.

**Strengths:**

1. The paper presents an adaptation of the shrink-and-perturb technique (originally from Ash & Adams, 2020) to the NCO domain. While the core SP method isn't novel, its application to bridge generalization and instance-specific optimization paradigms in unsupervised NCO shows effectiveness.

2. The experimental methodology is rigorous and comprehensive.

3. The paper is easy-to-follow and the motivation is clear.

**Weaknesses:**

1. The core contribution (parameter fusion via shrink-and-perturb) represents a relatively minor methodological advance. The methodology section is only one page long, and the core method can be expressed in a single line of equation, which is also borrowed from prior work. The paper would benefit from either deeper theoretical analysis of why SP works particularly well in the NCO context or extension to more sophisticated adaptation mechanisms.

2. While the paper claims "negligible additional computational cost", the tables show that TACO methods require the same computational time as fine-tuning baselines (which is substantially more than the base models without adaptation). The comparison should more clearly articulate that the overhead is negligible relative to the fine-tuning alternative, not relative to no adaptation.

**Questions:**

1. Could the authors provide more insight into why the optimization landscape around trained NCO parameters is "less conducive to rapid adaptation"? Is this specific to the unsupervised objectives used in NCO, or would this phenomenon occur with supervised NCO models as well?

2. The paper shows TACO works across different $\lambda_{shrink}$ values, but the optimal values differ significantly between EGN ($\lambda_{shrink}=0.3$) and Meta-EGN ($\lambda_{shrink}=0.7$). What explains this difference, and how should practitioners select these hyperparameters without access to validation data?

3. Have the authors considered other test-time adaptation techniques beyond shrink-and-perturb? For instance, could techniques like batch normalization adaptation or entropy minimization be effective in the unsupervised NCO setting?

---

### Official Review · Reviewer_MSos · 2025-10-27

**Soundness:** 3
**Presentation:** 2
**Contribution:** 2
**Rating:** 4
**Confidence:** 4

**Summary:**

In this article, the authors a model agnostic framework (coined as TACO) to address combinatorial optimization problems with unsupervised learning. This falls in the line of research started, among others, by Nikolaos Karalias and Andreas Loukas in 2020.

The main contributions are:

- a methodology for data-driven alternatives to classical solvers for combinatorial optimization

- an upgrade of standard fine tuning of their models, starting from a trained model, then adapted to specific instances

- Experiments demonstrating the performance of their method

**Strengths:**

- The problem of performance of data-driven techniques for various problems is a known and fundamental problem in the combinatorial optimization community. It is a relevant topic to study and explore.

- The shrink and perturb (SP) method in interesting.

- Extensive experimental results.

- The article can serve as a baseline and starting point for ideas to improve methods for data-driven techniques in combinatorial optimzation.

**Weaknesses:**

- I find the presentation of the method (Section 3) could be improved. For example, the explanation of Figure 1 is not entirely there: if the authors choose to put this figure in the main text, and in one of the main sections (the methodology), they should explain better what the EGN-rand-FT is next to it, and not drown in into the experimental details of Section 4.

- The conceptual and theoretical contributions are absent. Combined with the next weakness point, this becomes problematic.

- The experimental results are not very convincing: when TACO does outperform the other methods, it is only by a very fractional percentage, the practical gains are very unclear at this point. Maybe the other can clarify on this point.

**Questions:**

- Suggestion: bring changes to Section 3: move the figure in another Section, or describe and explain what is the message behind it.

- Can the authors confirm that the methodology Meta-EGN-FT precedes their work? In Table 1 this method seems to perform better than TACO on most of the problems (the problems where TACO is not outlined in grey).

---

### Official Review · Reviewer_VbRJ · 2025-10-28

**Soundness:** 2
**Presentation:** 2
**Contribution:** 2
**Rating:** 4
**Confidence:** 5

**Summary:**

This paper proposes a middle ground between two paradigms in neural combinatorial optimization (NCO), generalization-oriented and instance-specific solvers, by leveraging test-time scaling.
During test-time scaling, the method applies the Shrink-and-Perturb (SP) technique to control the initialization, enabling more efficient exploration.
As a result, it attains higher ApR than both Meta-EGN and EGN on a range of out-of-distribution datasets.

**Strengths:**

- The distinction between Meta-EGN and TACO is clear and intuitive, and the performance advantage over Meta-EGN is demonstrated quantitatively.
- It is novel and interesting to examine the intermediate regime between generalization and instance-specific optimization and to integrate the two.
- The perspective of designing algorithms that efficiently solve dynamic problems is compelling.

**Weaknesses:**

- **On novelty**: The method represents an incremental extension, combining EGN, originally developed for generalization, with test-time scaling. The effectiveness of the Gaussian perturbation remains unclear. In particular, its behavior on high-dimensional problems, the typical targets of CO heuristics, remains uncertain.
- **On Lines 47–48** (*this paradigm stays unaffected by distribution shifts and dynamic changes, but lacks the ability to generalize from broader patterns and is potentially susceptible to becoming trapped in poor local optima during optimization*): While it is true that this paradigm does not generalize, there exist highly parallelized approaches [1] that capture the problem distribution; coupled with fine-tuning, they can yield reasonably good solutions. Since that fine-tuning strategy is related to your TTS, it should at least be included in Related Work. Moreover, an annealing-based approach [2, 3] mitigates local-optimum issues via an annealing strategy; failing to discuss this is problematic.
- **On baselines**: The baselines should include methods such as CRA-PI-GNN [2] that are designed to avoid poor local optima. Because EGN and Meta-EGN focus on improving average performance, they may still yield relatively poor solutions in some cases, which could make TTS appear more beneficial. It is necessary to examine whether TTS performs appropriately when combined with a method such as CRA-PI-GNN, which actively avoids local minima and already yields reasonably good solutions.
- **On SP**: It is not convincing that Gaussian perturbations are effective for escaping local minima in high dimensions. Please provide evidence that Gaussian perturbations remain effective in high-dimensional settings. Specifically, does claim (iii), *helping escape poor local minima via stochastic perturbations* hold consistently in high dimensions?
- **On speed-quality trade-offs**: There should be a quantitative discussion comparing (i) using a generalized model to solve test instances quickly versus (ii) solving test instances efficiently in parallel [1]. Which approach is preferable in terms of runtime–performance trade-offs?
- **On sampling-based methods**: I would like to see comparisons with recent sampling-based methods [4, 5] that are lighter-weight yet achieve strong performance, sometimes surpassing ML based-approachs. These methods are lightweight and could potentially achieve better performance more quickly than your test-time scaling. In high-dimensional problems with many nodes, sampling-based methods may require fewer transition parameters and therefore run faster than TTS.

### References
- [1]: Yuma Ichikawa and Hiroaki Iwashita, *Continuous Parallel Relaxation for Finding Diverse Solutions in Combinatorial Optimization Problems*, Transactions on Machine Learning Research, 2025.
- [2]: Yuma Ichikawa, *Controlling Continuous Relaxation for Combinatorial Optimization*, NeurIPS2023.
- [3]: Haoran Sun et al., *Annealed Training for Combinatorial Optimization on Graphs*, arXiv preprint arXiv:2207.11542, 2022.
- [3]: Haoran Sun et al., *Revisiting Sampling for Combinatorial Optimization*, ICML2023.
- [4]: Yuma Ichikawa and Yamato Arai, *Optimization by Parallel Quasi-Quantum Annealing with Gradient-Based Sampling*, ICLR2025.

**Questions:**

- What advantages does your approach have over methods like CRA-PI-GNN that are designed not to become trapped in local minima?

---

### Note · Authors · 2025-11-22

I have read and agree with the venue's withdrawal policy on behalf of myself and my co-authors.